# Highly Pathogenic Avian Influenza H5N1 in Cats (*Felis catus*): A Systematic Review and Meta-Analysis

**DOI:** 10.3390/ani15101441

**Published:** 2025-05-16

**Authors:** D. Katterine Bonilla-Aldana, Jorge Luis Bonilla-Aldana, Jaime David Acosta-España, Alfonso J. Rodriguez-Morales

**Affiliations:** 1College of Medicine, Korea University, Seoul 02841, Republic of Korea; dbonilla@korea.ac.kr; 2Grupo de Virologia, Universidad El Bosque, Bogotá 111321, Colombia; jorg.bonilla@udla.edu.co; 3School of Medicine, Pontificia Universidad Católica del Ecuador, Quito 170525, Ecuador; 4Health Sciences Faculty, Universidad Internacional SEK (UISEK), Quito 170120, Ecuador; 5Institute of Microbiology, Friedrich Schiller University Jena, 07743 Jena, Germany; 6Centro de Investigación para la Salud en América Latina (CISeAL), Pontificia Universidad Católica del Ecuador, Quito 170120, Ecuador; 7Faculty of Health Sciences, Universidad Científica del Sur, Lima 15307, Peru; 8Grupo de Investigación Biomedicina, Faculty of Medicine, Fundación Universitaria Autónoma de las Américas-Institución Universitaria Visión de las Américas, Pereira 660003, Colombia

**Keywords:** H5N1, cats, highly pathogenic avian influenza, zoonosis, systematic review, meta-analysis, one health

## Abstract

Highly pathogenic avian influenza H5N1 is a dangerous virus that primarily affects birds but has recently been found in other animals, including domestic cats. This raises concerns because cats often live close to humans and other animals, potentially acting as a bridge for the virus to spread further. In this study, we reviewed and analysed global data to understand how often H5N1 has been found in cats and the consequences. Although cat infections are relatively rare, they are becoming more common, especially in recent years. Many infected cats became seriously ill or died, showing signs of severe lung and brain problems. These findings highlight the urgent need to monitor cats for H5N1 and adopt better prevention strategies to protect animals and people.

## 1. Introduction

Highly pathogenic avian influenza (HPAI) H5N1 is a zoonotic infection caused by a subtype of the influenza A virus [1,2,3,4,5]. First identified in Hong Kong in 1997, H5N1 primarily affects birds but has occasionally infected humans, often with high fatality rates [6,7,8,9]. The virus remains a significant global health concern due to its capacity to mutate and the ongoing risk of a potential pandemic [10,11]. Human infections typically result from direct or indirect contact with infected poultry. Recent outbreaks in animals, alongside sporadic human cases, highlight the urgent need for continuous surveillance, rapid response systems, and ongoing research into vaccines and antiviral treatments [12,13].

In recent years, H5N1 has been increasingly detected in a range of non-avian, non-human species, raising alarm over its ability to cross species barriers and adapt to new hosts [14,15,16]. Infections have been reported in various mammals, including foxes, mink, seals, sea lions, and even bears, typically through ingesting infected birds or exposure to contaminated environments. These cases are especially concerning in carnivorous and scavenging species. Remarkably, outbreaks among farmed mink have shown limited mammal-to-mammal transmission, suggesting the virus may be evolving further toward efficient transmission in mammals [17,18,19].

A particularly unusual and concerning development is the recent detection of H5N1 in cattle [20,21,22]. Infected dairy cows have exhibited mild symptoms, including reduced milk production and fever [15,16,23]. Since cattle have not traditionally been considered susceptible to avian influenza viruses, this unexpected host expansion raises critical questions about the virus’s adaptability and its potential for sustained transmission within livestock populations. In theory, their respiratory and gastrointestinal physiology and prior surveillance data suggested limited or no susceptibility to these viruses, making this apparent host jump both surprising and concerning. It also underscores the urgent need for surveillance and investigation into the mechanisms by which the virus infects and behaves in this novel host species. Feeding practices may also influence transmission risk, particularly in households or farms where pets are fed raw meat or unpasteurised dairy products. Raw beef or poultry from infected livestock may harbour the viable virus, potentially exposing companion animals such as cats and dogs. In contrast, properly cooked food is not considered a transmission risk due to the inactivation of H5N1 at standard cooking temperatures. Nevertheless, the increasing popularity of raw pet food diets raises concern and warrants further investigation and public awareness [24,25].

Notably, several outbreaks on dairy farms have involved simultaneous infections in domestic cats [26,27,28]. In such cases, it is believed that cats contracted H5N1 either by consuming raw milk or through exposure to contaminated farm environments [24,28,29]. The presence of H5N1 in both cattle and cats on the same farms suggests a previously under-recognised transmission route—from cattle to companion animals—which could pose additional risks for humans and animals alike [15,30,31].

This highlights the changing ecology of the virus and underscores the need for enhanced biosecurity measures on farms [29,32]. Because cats often interact closely with both livestock and humans, they may act as intermediate hosts, potentially facilitating viral adaptation or even zoonotic spillover [24,33]. Alarmingly, H5N1 infections in cats have been linked to severe disease and death, indicating that they are highly susceptible and may contribute to an increased viral load in affected environments [30,34,35].

Cats’ role in transmitting H5N1 warrants particular attention [27,28,36]. Infected cats can develop severe respiratory and neurological symptoms, frequently resulting in death [37,38]. Their infections have been linked to consuming infected birds or raw milk from cows confirmed to be H5N1-positive [15,24,39]. As companion animals in close contact with people, cats could act as bridges between animal hosts and humans, increasing the risk of cross-species transmission. This underscores the need for the vigilant surveillance of domestic animals, stricter biosecurity protocols in mixed-species settings, and increased public awareness to mitigate the growing public health threat posed by feline H5N1 infections [16,40].

This investigation wanted to assess the overall prevalence of H5N1 influenza infections in domestic cats (*Felis catus*) through a systematic review with a meta-analysis.

## 2. Methodology

The preparation of this study adhered to the methodological framework outlined in the PRISMA (Preferred Reporting Items for Systematic Reviews and Meta-Analyses) guidelines to ensure transparency and rigor in reporting (Figure 1) [41].

### 2.1. Sources of Information and Strategy of Search

An extensive scientific literature search was performed to locate studies reporting on H5N1 infection rates in domestic cats (*Felis catus*). Sources included PubMed, Scopus, and Web of Science. The search protocol’s development adhered to the PRESS (Peer Review of Electronic Search Strategies) guidelines to optimise the accuracy and comprehensiveness. All available records from the launch of each database through 1 March 2025 were considered, with no limitations placed on the publication language.

### 2.2. Selection of Studies and Extraction of Data

This review focused exclusively on observational research assessing the prevalence of H5N1 in domestic cats. Systematic reviews, scoping reviews, narrative reviews, and editorial letters were excluded from consideration. Case reports, however, were retained for an independent supplementary evaluation.

All identified records were imported into the Rayyan QCRI platform for data screening and organisation [42]. Duplicate records were identified and removed prior to screening. Titles and abstracts were then reviewed, followed by full-text assessments to determine eligibility based on predefined inclusion criteria. Studies that did not meet these criteria were excluded. Four reviewers conducted the screening process independently, with disagreements resolved collectively through discussion. Data extraction was performed using a standardised form developed in Microsoft^®^ Excel^®^, capturing key study details such as the author, publication year, geographic location, diagnostic method, study population, and the number of H5N1-positive cats.

### 2.3. Assessment of Bias Risk

To evaluate the risk of bias in the selected studies, the Newcastle–Ottawa Scale tailored for cross-sectional research was employed [43]. Two reviewers independently conducted the assessments. Studies scoring seven or more stars were classified as having a low risk of bias, while those receiving six stars or fewer were considered to carry a high risk.

### 2.4. Publication Bias Assessment

Current methodological guidance advises against assessing publication bias in proportional meta-analyses. Traditional approaches—such as funnel plots and Egger’s test—are considered unsuitable for this context. These methods are based on the assumption that studies with positive findings are more likely to be published than those with negative outcomes. However, there is no universally accepted definition of a ‘positive’ result in prevalence-based analyses. Moreover, there is insufficient evidence to support that proportion estimates behave reliably when subjected to these bias detection techniques [44,45].

### 2.5. Data Analysis

A quantitative data analysis was performed using Stata version 16. A random-effects meta-analysis was applied based on the DerSimonian and Laird approach to account for expected variability across studies. The Clopper–Pearson method was used to calculate 95% confidence intervals. The Freeman–Tukey double arcsine transformation served to stabilise the variance in proportion estimates. The between-study heterogeneity was evaluated using Cochran’s Q test and the I^2^ statistic, with I^2^ values ≥ 60% interpreted as substantial heterogeneity and Q test significance defined at *p* < 0.05. Subgroup analyses were stratified by the diagnostic method and geographical region. Additionally, a sensitivity analysis was conducted by excluding studies with a high risk of bias.

## 3. Findings

### 3.1. Selection of Studies

Our search strategy yielded 3182 records across the combined databases. After removing duplicates and screening for titles and abstracts, 199 articles underwent a full-text review. Finally, 21 articles were included in the systematic review and meta-analysis, 8 of which were used for quantitative synthesis [46,47,48,49,50,51,52,53,54,55,56,57,58,59,60,61,62,63,64,65,66,67] (Table 1). The PRISMA flow chart is shown in Figure 1.

### 3.2. Characteristics of Included Studies

The characteristics of the included articles are summarised in Table 1. A total of eight prevalence or seroprevalence articles were included, evaluating 3586 cats. The studies spanned from 2008 to 2024; however, in 2015, there were three (Table 1). The studies were distributed as follows: Asia, with four studies (three from China and one from Indonesia); Europe, with three studies (from Germany, Italy, and Poland); and one study from Africa (Egypt) (Table 1). No studies from other regions were included. Three studies included domestic cats, three included stray cats, and one included cats living near H5N1-positive poultry farms (Table 1). Five studies assessed the samples using a RT-PCR by pharyngeal or nasal swabs (1666 cats), six studies employed the hemagglutination inhibition (HI) assay from serum (3058 cats), two studies used the neutralisation (NT) assay from serum (1720 cats), and one study used an ELISA from serum (279 cats) (Table 1). Only one study additionally used genome sequencing and phylogeny (Table 1).

### 3.3. Risk of Bias Assessment

In the risk of bias assessment, seven studies were at high risk, while the remaining were at low risk of bias. There is no substantial evidence of publication bias (Egger’s test, −9.44; *p* = 0.1791), although the test may be underpowered due to the small number of studies.

### 3.4. Prevalence/Seroprevalence of H5N1 in Cats by Any Method

The global pooled prevalence of H5N1 in cats, as determined by any method, assuming immunological methods that allow for the suggestion of infection, in addition to RT-PCR, was 0.7% (95% CI: 0.3–1.1%), with a high heterogeneity (I^2^ = 86.478%, τ^2^ < 0.001, Q^2^ = 110.934) (Figure 2) (Table 2).

### 3.5. Prevalence/Seroprevalence of H5N1 in Cats by Molecular and Immunological Methods

Regarding the methods, the prevalence of H5N1 infections as determined by the RT-PCR was 0.8% (95% CI, 0.0–2.2%), with high heterogeneity (I^2^ = 93.187%, τ^2^ < 0.001, Q^2^ = 58.710) (Figure 3). The prevalence of H5N1 infections as determined by the HI assay was 1.5% (95% CI, 0.5–2.4%), with high heterogeneity (I^2^ = 83.89%, τ^2^ < 0.001, Q^2^ = 43.452) (Figure 4). The prevalence of H5N1 infection as determined by the NT assay was 0.3% (95% CI, 0.0–0.5%), with nonassessable heterogeneity (I^2^ = 0%, τ^2^ < 0.001, Q^2^ = 0.674) (Figure 5) (Table 2).

### 3.6. Prevalence/Seroprevalence of H5N1 in Cats by Type of Cats

When analysed by subgroups, domestic cats had a higher prevalence, 20.0% (95% CI, 0.0–41.2%) (Figure 6), compared to other types of cats: stray, 2.3% (95% CI, 0.0–5.4%), shelter cats, 0.3% (95% CI, 0.0–0.7%), and farm cats, 0.3% (95% CI, 0.0–1.1%) (Figure 6) (Table 2).

### 3.7. Prevalence/Seroprevalence of H5N1 in Cats by Geographical Regions

Grouping geographically by regions, the prevalence was higher in Africa, 32.0% (95% CI, 13.7–50.3%) (Figure 7), compared to other regions: Europe, 2.7% (95% CI, 0.0–6.0%) and Asia, 2.0% (95% CI, 0.1–3.8%) (Figure 7) (Table 2).

### 3.8. Prevalence/Seroprevalence of H5N1 in Cats by Years and Periods

Analysing by years, it was observed the prevalence was higher in 2023, 54.3% (95% CI, 40.0–68.7%) (Figure 8), and lower in 2008, 0.3% (95% CI, 0.5–1.1%), and 2024, 0.3% (95% CI, 0.0–0.7%) (Figure 8). However, a grouping of periods revealed a higher prevalence after 2015, at 19.4% (95% CI, 0.0–41.6%), compared to the period up to 2015, at 1.5% (95% CI, 0.1–2.9%) (Figure 9) (Table 2).

### 3.9. Molecular and Serological Findings from Case Reports

From the case series and reports (13 publications), we found 35 cats infected with H5N1 influenza (Table 3). The diagnosis was confirmed by a RT-PCR in most cats (97%); additionally, genome sequencing was performed in 46% of the cats, identifying the clade 2.3.4.4b in all of them (Table 3). Other techniques used complementarily were the HI assay (six cats), the NT assay (three cats), the ELISA (three cats), and viral isolation (three cats).

Reports were made between 2006 and 2025; interestingly, 59% of the cases were reported in 2024 and 2025 (Table 3). Cases were reported mainly in the USA (43%), followed by the Republic of Korea (26%), Poland (14%), Germany (9%), France (3%), Italy (3%), and Thailand (3%) (Table 3). No reports from Africa or Latin America were found. Age was available in 10 of the cases (29%); among them, the median age was 1.9 years, ranging from 0.5 to 13 years (IQR, 0.6–5.0 years) (Table 3). Only 12 cases reported sex (34%), corresponding to eight males (23%) and four females (11%). Among 80% of the cases, an outcome was reported, corresponding to 74% fatal outcomes (these included four cases of euthanasia) and 6% survival (Table 3). The median time to death after the onset of illness (days) among those where this was reported (n = 13) was 2 days, ranging from 1 to 10 days (IQR, 2–4 days).

Most of the cats presented with clinical manifestations (97%) (Table 4), exhibiting a high diversity of 35 different symptoms, primarily lethargy (29%), ataxia (23%), progressive neurological deterioration (23%), and fever (20%) (Table 4).

From the total, necropsies were performed in 18 cats (34%), exhibiting a high diversity of 28 different findings, affecting especially the lung, liver and brain, primarily with interstitial pneumonia (11/18, 61%), liver necrosis foci (5/18, 28%), pulmonary oedema (5/18, 28%), and multifocal encephalitis (5/18, 28%) among others (Table 5). Table 6 details the pathological findings reported in these deceased cats.

The reported treatment for five cats included fluid therapy (four cats), dexamethasone (two cats), maropitant (two cats), aspirin, atropine, amoxicillin–clavulanic acid, enrofloxacin, tolfenamic acid, gentamicin, ampicillin, ceftriaxone, topical emodepside/praziquantel, metronidazole, and meloxicam (one per cat).

## 4. Discussion

This systematic review and meta-analysis provide the first comprehensive quantitative data synthesis regarding H5N1 infection in cats (*Felis catus*) across different regions and populations. The overall pooled prevalence of H5N1 infections in cats was 0.7% (95% CI: 0.3–1.1%), with considerable heterogeneity. Although the prevalence appears relatively low on a global scale, the detection of infections across domestic, stray, shelter, and farm cat populations, coupled with high mortality rates in some outbreaks, underscores the growing public health significance of feline infections with HPAI H5N1 [11,14,21,24,28,34,37,38,46].

The overall prevalence reported in this study aligns with earlier observations in individual settings but highlights several critical patterns [1,12,17,18,24,28,29,38]. Notably, domestic cats had the highest estimated prevalence at 20.0% (95% CI: 0.0–41.2%), compared to lower rates among stray (2.3%), shelter (0.3%), and farm (0.3%) cats [17,24,26,28,29,34,36,37,47,48,49,50,51]. While these wide confidence intervals reflect the variability and sample size limitations, the findings suggest that cats living in close contact with humans may be at a heightened risk of acquiring H5N1 infections, potentially through the ingestion of infected animal products, such as raw poultry or milk from infected dairy cattle, or through exposure to contaminated farm environments [15,16,20,21,22,24,25,39]. Domestic cats may ultimately come into contact with infected wild birds, posing a potential risk due to their migratory behaviours [52]. Also, the more feasible access of domestic cats can influence such a higher prevalence. Sampling and/or collecting data from a cat population with restricted freedom and a well-defined tutor is less complicated than doing so in another category of populations (stray, shelter, and farm).

Geographical differences in prevalence were also marked, with the highest pooled prevalence observed in Africa (32.0%), followed by Europe (2.7%) and Asia (2.0%). However, this disparity may be partially attributed to the small number of African studies (n = 1), which warrants caution in interpretation. Although based on a limited dataset, the African finding highlights the importance of regional surveillance in underrepresented areas, mainly where informal farming practices may increase cross-species contact and transmission risks [2,11,31,53]. Another aspect that warrants discussion is that no studies have been conducted in the Americas. However, according to reports published by the Pan American Health Organization (PAHO) in 2025 (https://www.paho.org/sites/default/files/2025-01/2025-jan-24-phe-epiupdate-avian-influenza-eng-final.pdf, accessed on 30 January 2025) and the U.S. Department of Agriculture (USDA) in 2024 (https://www.aphis.usda.gov/livestock-poultry-disease/avian/avian-influenza/hpai-detections/mammals, accessed on 30 January 2025), the avian influenza (H5N1) virus has been detected in the USA and Canada. Such reports are not published studies, just epidemiological reports, and, more importantly, they are not in cats.

Temporally, the prevalence appears to have increased substantially in recent years. Data from 2023 indicated a markedly elevated pooled prevalence of 54.3% (95% CI: 40.0–68.7%) [5,7,11,12,19,22,23,29,34,40,48]. Similarly, the subgroup analysis by period demonstrated a notable rise in prevalence after 2015 (19.4%) compared to the earlier period (1.5%). This upward trend may reflect an actual increase in the viral transmission to cats, driven by the ongoing global expansion of H5N1 clade 2.3.4.4b and its broader host range, including the recent and concerning detection of infections in dairy cattle. Alternatively, it may also result from the improved surveillance, diagnostic capacity, and growing awareness of feline cases [24,26,29,34,38,46,47,48,49,50,51,54,55,56,57].

These findings carry profound implications for One Health approaches, given the emerging role of cats as potential intermediate or incidental hosts [11,31]. While a sustained cat-to-cat or cat-to-human transmission has not yet been documented, the susceptibility of cats to severe disease and their proximity to humans raise concerns [27]. In several outbreaks, cats exhibited severe respiratory and neurological signs, and post-mortem analyses revealed neurotropism and a widespread viral distribution, suggesting systemic involvement and potentially high environmental viral shedding [34]. These features may contribute to an amplifying viral circulation in shared spaces with humans, livestock, and wildlife [17].

Methodologically, the variation in prevalence estimates according to diagnostic methods was expected. The serological HI assay produced the highest pooled prevalence at 1.5%, followed by the RT-PCR (0.8%) and NT assay (0.3%). These differences may reflect the timing of the sample collection relative to infection, as the PCR detects current infections, while serological methods may identify prior exposure. Nonetheless, the consistent detection of infection across all methods reinforces the conclusion that feline infections are not sporadic anomalies but part of the evolving epidemiology of H5N1 [6,31,53,58].

High heterogeneity was observed across all subgroup analyses except for the NT assay group. This suggests variability in study designs, populations, geographic contexts, and assay sensitivities. Notably, the risk of bias assessment found that most studies were at high risk of bias. While the meta-analysis incorporated a sensitivity analysis and adhered to rigorous selection and statistical criteria, the quality and consistency of the primary data remain limitations.

A particularly striking finding is the sharp spike in prevalence in recent years. The 2023 data from Poland, where an outbreak of H5N1 was confirmed in multiple domestic cats, raise significant concerns. Genome sequencing confirmed clade 2.3.4.4b, as found in infected birds and other mammals, suggesting a possible exposure to everyday environmental sources or direct interspecies transmission [27,49,55]. These incidents suggest that cats, particularly those fed raw poultry or milk, may serve as indicators of broader epizootic activity and could play a role in viral adaptation or reassortment [32,34].

This emerging evidence highlights the need for a renewed emphasis on several urgent areas of intervention. First, veterinary and public health surveillance systems must be expanded to routinely include companion animals, particularly in outbreak-prone settings or where livestock and wildlife intersect [59,60]. Second, guidelines on biosecurity and feeding practices should be updated to include warnings against feeding cats raw poultry or unpasteurised milk, especially in areas with confirmed avian influenza activity. Third, the diagnostic testing capacity should be expanded for both symptomatic and asymptomatic cats in outbreak investigations to better assess the extent of the viral circulation and potential reservoirs [30,61,62].

Moreover, the growing number of H5N1-infected mammals, including foxes, mink, sea lions, and now cats and cattle, highlights the virus’s increasing adaptability and raises concerns about its pandemic potential [17,18,19]. While there is no definitive evidence of a sustained mammal-to-mammal transmission, the virus’s continuing mutations in mammalian hosts could enhance its zoonotic risk. Monitoring such mutations, particularly in genes associated with receptor binding, polymerase activity, or immune evasion, is critical [63,64].

The compilation of case reports and series provides compelling evidence of the ongoing and possibly intensifying zoonotic threat posed by highly pathogenic avian influenza H5N1 in domestic and shelter cats. With 35 confirmed cases across 13 publications, this analysis highlights the evolving epidemiological, clinical, and pathological profile of feline H5N1 infections, with notable implications for One Health surveillance and public health [11,29,35,59,62].

A significant epidemiological finding is the temporal concentration of cases, with 59% reported in 2024 and 2025. This suggests a marked resurgence or improved detection of feline H5N1 infections in recent years. The predominance of cases in the United States and the Republic of Korea, together accounting for nearly 70% of reports, may reflect regional outbreaks linked to contaminated poultry, raw milk, or exposure to infected wild birds and mammals [20,24,25,38,46]. Interestingly, no studies were found from Sub-Saharan Africa or Latin America, which may indicate either underreporting or a lower surveillance capacity in these regions [11]. The overrepresentation of shelter cats further suggests high-risk environments where exposure to raw animal products and close contact with other animals are daily occurrences [38,54,61,65].

From a diagnostic standpoint, molecular confirmation via a RT-PCR was nearly universal (97%), and genome sequencing was performed in 46% of cases, consistently identifying the clade 2.3.4.4b [7,23,24,28,34,35,40,48,50,55,66]. The widespread identification of this clade across different geographic settings aligns with previous findings about its global dissemination and mammalian adaptation. Serological techniques, such as HI and NT assays, were less frequently employed, highlighting a diagnostic reliance on nucleic acid-based methods during acute presentations [23,40].

Clinically, H5N1 infections in cats exhibited high morbidity and mortality. A striking 74% of cases resulted in death, and most succumbed within a median of two days following symptom onset [6,13,24,27,32,49,53]. The rapid disease progression highlights the virulence of the virus in felines, particularly those infected with clade 2.3.4.4b, which is known for its enhanced neurotropism [15,22,27,29,32,34,50,64]. Among symptomatic cats (97%), neurological signs (ataxia and progressive deterioration) and systemic signs (lethargy and fever) predominated [15,22,27,29,32,34,50,64]. Such manifestations mirror the pathogenesis observed in other mammalian hosts [17,18,19], suggesting a systemic viral dissemination and the involvement of the central nervous system [14,15,16].

The necropsy and histopathological findings further elucidate the multisystemic nature of H5N1 infections in cats. Interstitial pneumonia (61%) and multifocal encephalitis (28%) were the most frequent lesions, reflecting a predominant respiratory and neurological involvement [15,17,22,47,64]. The diversity of the necropsy findings—including liver necrosis, pulmonary oedema, myocarditis, and chorioretinitis—underscores the virus’s broad tissue tropism [13,27,34,39,50]. Particularly notable is the high prevalence of central nervous system lesions, which included nonsuppurative encephalitis, neuronal necrosis, and perivascular inflammatory infiltrates, sometimes accompanied by meningeal haemorrhages and malacia. Such findings are consistent with the neurotropic behaviour of clade 2.3.4.4b and parallel those observed in experimentally infected mammals and natural spillovers in wild carnivores [50,67,68].

The source of infection, in many cases, remains speculative but often involves plausible zoonotic pathways [1,31,65,69]. Documented exposures include the consumption of raw poultry, infected milk from H5N1-positive cattle, and contact with infected wild birds or contaminated environments [15,16,23,24,25,29,30,32,33,39]. These routes emphasise the potential for interspecies transmission, particularly when human–animal interfaces are breached, such as through feeding practices or shared habitats in shelters and farms. The recurrence of cases associated with raw milk and poultry points to a need for stricter control measures in animal product handling and feeding, especially amid avian influenza outbreaks in livestock [15,16,23,24,25,29,30,32,33,39].

Therapeutic attempts were limited and generally ineffective, with most treated animals still succumbing to the disease. This reflects both the aggressiveness of the infection and the lack of standardised antiviral protocols for veterinary use against H5N1. Supportive treatments, such as fluid therapy and corticosteroids, were the most common, but none appeared to alter the clinical course significantly [5,12,15,27].

Finally, the case series analysis underscores that cats are susceptible to severe and often fatal H5N1 infections, particularly from the clade 2.3.4.4b lineage. The high prevalence of neurological and respiratory signs, a rapid progression to death, and extensive necropsy abnormalities highlight the need for increased vigilance and an integrated One Health surveillance [11,15,30,59,62]. Given the virus’s demonstrated zoonotic potential and its continued evolution in mammalian hosts, including domestic cats, the proactive monitoring of feline populations, especially in outbreak areas, is essential to mitigate the risk of spillover to humans and other animals [17,29,35,62].

## 5. Limitations

This study has several limitations. First, the small number of included studies and their uneven geographic distribution limit the generalisability of the findings. Most data came from Asia and Europe, with only one study from Africa and none from the Americas. Second, the methodological heterogeneity, including differences in diagnostic techniques and cat populations, contributed to the high statistical heterogeneity. Third, most studies were at a high risk of bias, which could affect the pooled estimates. Finally, the temporal variation in sampling periods may not accurately reflect the current prevalence, especially in light of recent outbreaks, underscoring the need for updated and regionally diverse surveillance data. Finally, we did not include any additional bibliographical databases.

## 6. Conclusions

This systematic review and meta-analysis highlights the emerging role of cats in the ecology of highly pathogenic avian influenza (H5N1). While the overall pooled prevalence in feline populations remains low, the detection of clade 2.3.4.4b infections—often with severe neurological and respiratory manifestations and high mortality—highlights their susceptibility and potential role in viral transmission dynamics. Domestic and shelter cats are at a heightened risk, especially those exposed to raw poultry or milk from infected cattle. The increasing number of confirmed cases in recent years, particularly in 2024 and 2025, suggests either an expanding epizootic or improved detection methods. Given their proximity to humans and other animals, cats may serve as sentinels or intermediate hosts. Enhanced surveillance, diagnostic vigilance, and integrated One Health strategies are urgently needed to monitor H5N1 in companion animals and mitigate the risks of zoonotic spillover. These findings underscore the importance of implementing biosecurity measures and enhancing public health preparedness in mitigating cross-species influenza threats.

## Figures and Tables

**Figure 1 animals-15-01441-f001:**
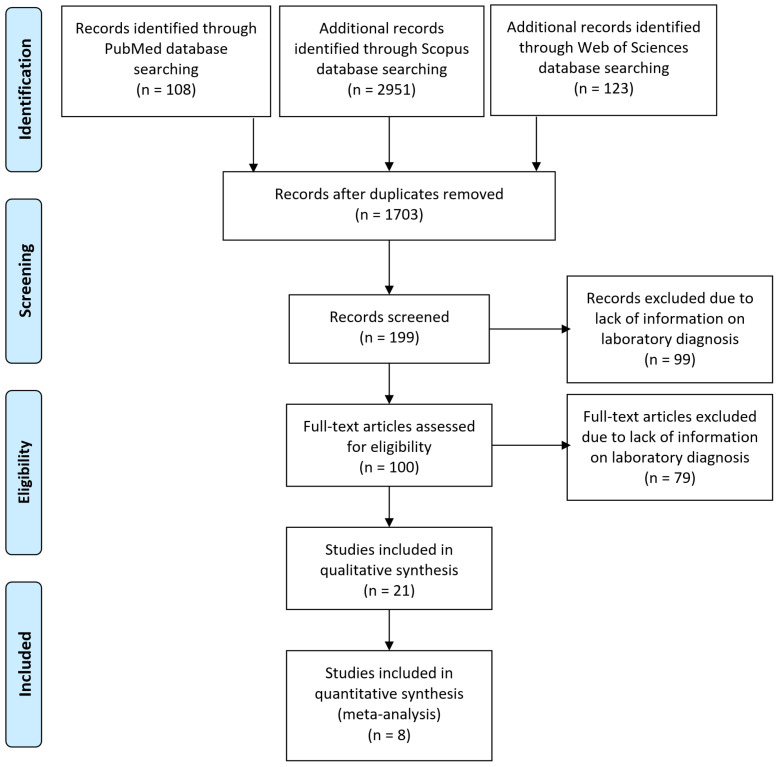
PRISMA diagram flow.

**Figure 2 animals-15-01441-f002:**
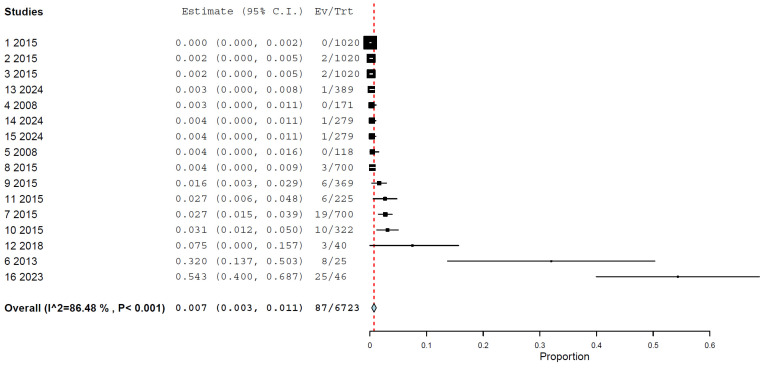
Prevalence of H5N1 in cats, regardless of method used.

**Figure 3 animals-15-01441-f003:**
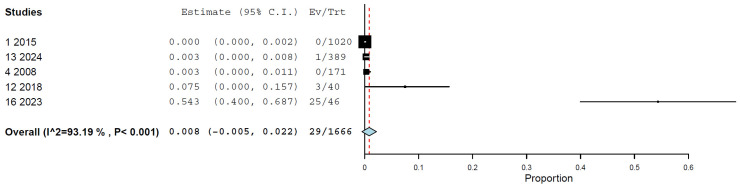
Prevalence of H5N1 in cats by RT-PCR.

**Figure 4 animals-15-01441-f004:**
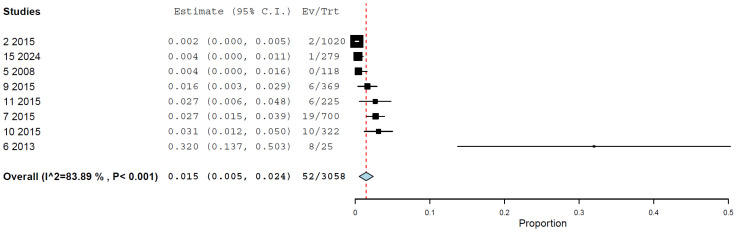
Prevalence of H5N1 in cats by HI assay.

**Figure 5 animals-15-01441-f005:**
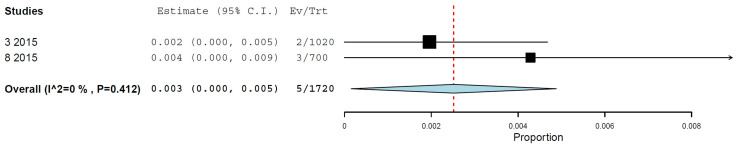
Prevalence of H5N1 in cats by NT assay.

**Figure 6 animals-15-01441-f006:**
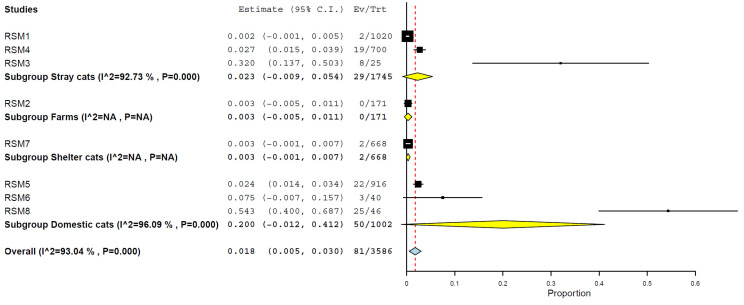
Prevalence of H5N1 in cats by type of cat.

**Figure 7 animals-15-01441-f007:**
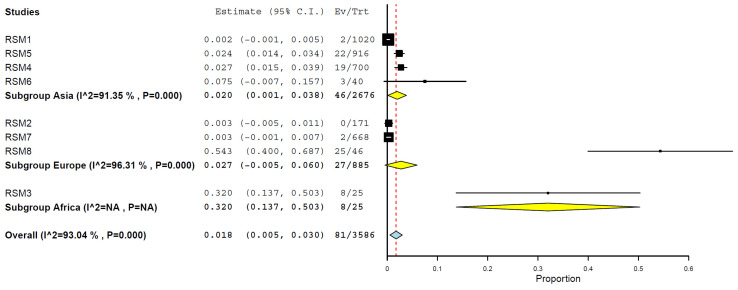
Prevalence of H5N1 in cats by geographic regions.

**Figure 8 animals-15-01441-f008:**
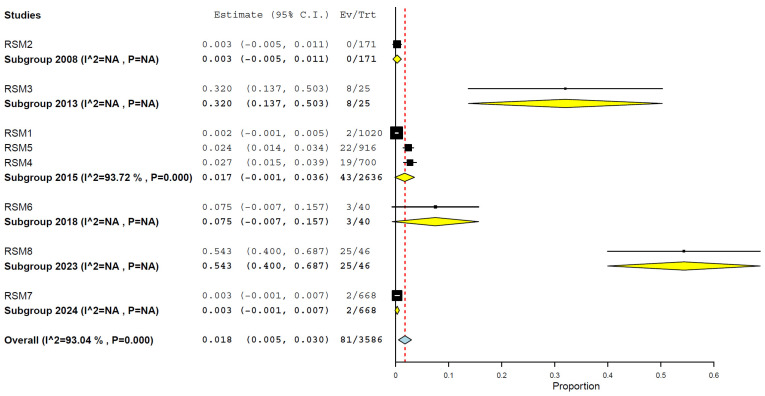
Prevalence of H5N1 in cats by years (2008–2024).

**Figure 9 animals-15-01441-f009:**
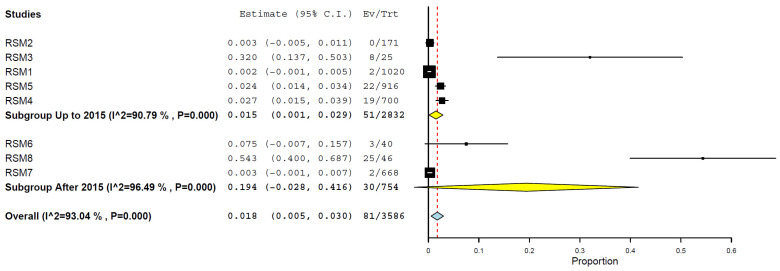
Prevalence of H5N1 in cats up to and after 2015.

**Table 1 animals-15-01441-t001:** Characteristics of the included studies.

Study	Publication Year	Study Years	Country	Study Type	N	Population	Test Employed	Reference
Prevalence of influenza A H5N1 virus in cats from areas with the occurrence of highly pathogenic avian influenza in birds	2008	2006–2007	Germany	Prevalence	171	Cats living near H5N1-positive poultry farms	RT-PCR, HI assay	[56]
Sero-prevalence of Avian Influenza in Animals and Humans in Egypt	2013		Egypt	Prevalence	25	Stray cats	HI assay	[53]
Detection Prevalence of H5N1 Avian Influenza Virus Among Stray Cats in Eastern China *	2015	2010–2011	China	Prevalence	1020	Stray cats	RT-PCR, HI assay, and NT assay	[54]
Serological evidence of avian influenza virus and canine influenza virus infections among stray cats in live poultry markets, China	2015		China	Seroprevalence	700	Stray cats	HI assay and NT assay	[57]
Sparse serological evidence of H5N1 avian influenza virus infections in domestic cats, northeastern China	2015	2013	China	Seroprevalence	916	Domestic cats	HI assay	[51]
Surveillance of Avian Influenza Virus of H5N1 Subtype in Backyard Animals and Its Introduction in Bali, Indonesia	2018	2005–2006	Indonesia	Prevalence	40	Domestic cats	RT-PCR	[58]
Outbreak of highly pathogenic avian influenza A(H5N1) clade 2.3.4.4b virus in cats, Poland, June to July 2023 **	2023	2023	Poland	Prevalence	46	Domestic cats	RT-PCR, Genome sequencing, and phylogeny	[55]
Influenza in feral cat populations: insights from a study in North-East Italy	2024	2021–2022	Italy	Prevalence	668	Shelter cats	RT-PCR, HI assay	[65]

HI assay, hemagglutination inhibition assay. NT assay, neutralisation assay. RT-PCR, reverse transcriptase polymerase chain reaction. * They used the virus A/duck/Guangdong/1/2013(H5N1) belonging to Clade 2.3.2. ** Complete genome sequences of 19 HPAI H5N1 virus-positive cats indicate that the viruses belonged to clade 2.3.4.4b, genotype CH (H5N1-A/Eurasian_Wigeon/Netherlands/3/2022-like).

**Table 2 animals-15-01441-t002:** Meta-analysis outcomes (random-effects model).

Meta-Analysis	Number of Studies	N	Pool Prevalence (%)	95% Confidence Interval	I^2 ‡^	*p*
All studies	8	3586	0.7	0.3–1.1%	86.48%	<0.001
By RT-PCR	5	1666	0.8	0.0–2.2%	93.19%	<0.001
By hemagglutination inhibition assay	6	3058	1.5	0.5–2.4%	83.89%	<0.001
By neutralisation assay	2	1720	0.3	0.0–0.5%	0.0%	0.412
Domestic cats	3	1002	20.0	0.0–41.2%	96.09%	<0.001
Stray cats	3	1745	2.3	0.0–5.4%	92.73%	<0.001
Shelter cats	1	668	0.3	0.0–0.7%	n/a	n/a
Farm cats	1	171	0.3	0.0–1.1%	n/a	n/a
Africa	1	25	32.0	13.7–50.3%	n/a	n/a
Europe	3	885	2.7	0.0–6.0%	96.31%	<0.001
Asia	4	2676	2.0	0.1–3.8%	91.35%	<0.001
2016–2024	3	754	19.4	0.0–41.6%	96.49%	<0.001
2008–2015	5	2832	1.5	0.1–2.9%	90.79%	<0.001

^‡^ I^2^ index for the degree of heterogeneity (percentage). RT-PCR, reverse transcriptase polymerase chain reaction. Some studies assessed simultaneous variables. Multiple studies assessed the prevalence using different methods. n/a, not applicable.

**Table 3 animals-15-01441-t003:** Characteristics of the cats in the case series and reports.

Cat No.	Year	Country	Type of Cat	Age, Years	Sex	Outcome	Comments	Ref.
1	2006	Thailand	Domestic	2	Male	Dead	The cat had eaten a pigeon 5 days before its death.	[37]
2	2007	Germany	Shelter	N/R	Male	Dead	They were found dead on the island of Rügen, approximately 1.6 km from sites where wild birds infected with highly pathogenic avian influenza virus (HPAIV H5N1), mainly swans and whooper swans as well as several species of geese and ducks, had been continuously detected during the previous three weeks. Note: They had access to the outdoors. Cat 2: found approximately 4 miles away from Cat 1. Cat 3: found 0.5 miles from the site where Cat 1 had been detected.	[47]
3				N/R	Male	Dead		
4				N/R	Male	Dead		
5	2023	Poland	Domestic	6	Male	Dead	Access to the outside, the diet included raw chicken meat, and the animal was vaccinated.	[27]
6	2023	Italy	Domestic	N/R	N/R	Unknown	Outbreak in birds within the same farm.	[66]
7	2023	France	Shelter	N/R	N/R	Unknown	A highly pathogenic H5N1 virus infected a domestic cat near a French duck farm, showing mammalian adaptation mutations and highlighting public health concerns. Diagnosed with RT-PCR, sequencing clade 2.3.4.4b.	[28]
8	2023	USA	Shelter	N/R	Male	Dead	He was brought to a clinic in northeast Nebraska because he was unable to walk and was anorexic. He had been kept outdoors and was missing for several days before being found in an abnormal condition. Three other free-roaming outdoor cats were housed at the facility. Diagnosed with RT-PCR, sequencing clade 2.3.4.4b.	[50]
9				0.5	Female	Dead	A cat from the same residence presented clinical signs 24 h after case 1. It was diagnosed with RT-PCR, sequencing clade 2.3.4.4b.	
10				0.7	Female	Dead	Difficulty walking, lethargy, and anorexia. This animal lived outdoors and was one of nine cats on the premises. Diagnosed with RT-PCR, sequencing clade 2.3.4.4b.	
11	2023	Poland	Domestic	N/R	N/R	Survived	A fatal H5N1 outbreak in Polish cats revealed viral mammalian adaptations, likely transmitted via contaminated poultry meat. Most infected cats died, raising public health and zoonotic transmission concerns.	[49]
12	2023	Poland	Domestic	N/R	N/R	Dead		
13	2023	Poland	Domestic	N/R	N/R	Survived		
14	2023	Poland	Domestic	N/R	N/R	Dead		
15	2024	Republic of Korea	Shelter	N/R	N/R	Dead	Respiratory and neurological symptoms.	[46]
16	2024	Republic of Korea	Shelter	N/R	N/R	Dead	In July 2023, H5N1 outbreaks occurred in two cat shelters in Seoul, resulting in high mortality rates linked to contaminated raw duck meat. At least 39 cats died; others were infected or seropositive. The virus exhibited mammalian adaptations, underscoring concerns about zoonotic transmission and food safety. This case series analysed 7 of those confirmed H5N1 infection in cats.	[38]
17	2024	Republic of Korea	Shelter	N/R	N/R	Dead		
18	2024	Republic of Korea	Shelter	N/R	N/R	Dead		
19	2024	Republic of Korea	Shelter	N/R	N/R	Dead		
20	2024	Republic of Korea	Shelter	N/R	N/R	Unknown		
21	2024	Republic of Korea	Shelter	N/R	N/R	Unknown		
22	2024	Republic of Korea	Shelter	N/R	N/R	Unknown		
23	2024	Republic of Korea	Shelter	N/R	N/R	Dead		
24	2024	USA	Shelter	Adults	Female	Dead	The cats were found dead with no apparent signs of injury and belonged to a resident population of approximately 24 domestic cats that had been fed milk from diseased cows. RT-PCR diagnosed both, sequencing clade 2.3.4.4b.	[24]
25	2024	USA	Shelter	Adults	Male	Dead		
26	2025	USA	Domestic	5	Female	Dead	Household 1: Lives exclusively inside the house. The owner worked with cattle. Diagnosed by RT-PCR, sequencing clade 2.3.4.4b, genotype B3.13.	[29]
27	2025	USA	Domestic	0.5	Male	Unknown	Household 2: Six days after the cat case, the cat was raised indoors. Diagnosed by RT-PCR, sequencing clade 2.3.4.4b, genotype B3.13.	
28	2025	USA	Stray	1.5	N/R	Dead	H5N1 clade 2.3.4.4b caused fatal neurotropic infections in cats, with high brain viral loads, unique mutations, and potential adaptation suggesting zoonotic and reassortment risks. RT-PCR diagnosed both, sequencing clade 2.3.4.4b.	[34]
29	2025	USA	Stray	0.5	N/R	Dead		
30	2025	USA	Domestic	0.5–2	N/R	Dead	They were fed mice, rats, and birds and offered raw and pasteurised colostrum and milk from diseased cows, which was diverted from bulk commercial milk. RT-PCR diagnosed them, sequencing for H5 and H5 2.3.4.4b in the three cats.	[48]
31	2025	USA	Domestic	0.5–2	N/R	Dead		
32	2025	USA	Domestic	0.5–2	Female	Dead		
33	2025	USA	Domestic	0.5–2	N/R	Dead		
34	2025	USA	Domestic	5	N/R	Unknown	Indoor cats reportedly had no contact with people outside the dairy farm where the outbreak occurred. RT-PCR diagnosed them, sequencing for H5 and H5 2.3.4.4b in the three cats.	
35	2025	USA	Domestic	13	N/R	Dead		

Ref, references. N/R, not reported.

**Table 4 animals-15-01441-t004:** Main clinical manifestations in the case reports of cats with influenza H5N1 (n = 35).

Systems	Manifestations	n	%
	Asymptomatic	1	3
	Symptomatic	34	97
Neurological	Lethargy	10	29
	Ataxia	8	23
	Progressive neurological deterioration	8	23
	Depressed	6	17
	Inability to get up	4	11
	Reduction in the threat reflex	4	11
	Blindness	4	11
	Cranial nerve abnormality	2	6
	Anisocoria	2	6
	Hypersensitivity	2	6
	Hiding behaviour	1	3
	Sleepiness	1	3
	Seizures	1	3
	Disorientation	1	3
	Nystagmus	1	3
	Tremors	1	3
Systemic	Fever	7	20
	Weakness	4	11
	Dehydration	3	9
	Panting	1	3
	Tachycardia	1	3
	Dyspnoea	1	3
	Hypothermia	1	3
	Shivering	1	3
	Sialorrhea	1	3
Ocular	Eye and nose discharge	6	17
	Inflammation of the eyes and nose	1	3
	Runny nose	1	3
	Purulent-watery ocular discharge	1	3
	Conjunctival injection	1	3
Enteric	Loss of appetite	5	14
	Anorexia	3	9
	Reduced appetite	2	6
Respiratory	Tachypnea	5	14
Miscellaneous	Jaw inflammation	1	3

**Table 5 animals-15-01441-t005:** Main necropsy findings in the case reports of cats infected with influenza H5N1 (n = 18).

Organ	Finding	n	%
Lung	Interstitial pneumonia	11	61
	Pulmonary oedema	5	28
	Lung congestion	2	11
	Bronchitis	2	11
	Bronchiolitis	2	11
	Lung necrosis	2	11
	Pulmonary hyperemia	2	11
	Lung perivasculitis	2	11
	Lung interstitial haemorrhage	1	6
Liver	Liver necrosis foci	5	28
	Liver inflammatory infiltrates	3	17
Neurological	Multifocal encephalitis	5	28
	Neuronal necrosis	3	17
	Necrotising meningoencephalitis	2	11
	Meningitis	2	11
	Brain perivascular infiltrate	1	6
	Brain congestion	1	6
Heart	Heart vasculitis	2	11
	Necrotising myocarditis	2	11
Multisystemic	Necrosis foci	1	6
	Multifocal necrosis	1	6
Spleen	Splenomegaly	1	6
	Spleen Congestion	1	6
Kidney	Kidney congestion	1	6
	Necrotic lesions and inflammation of the mesenteric plexus	1	6
Intestine/Mesenteric	Intestinal serous haemorrhage	1	6
	Chorioretinitis	1	6
Eye	Conjunctivitis	1	6

**Table 6 animals-15-01441-t006:** Detailed pathological findings in the case reports of cats infected with influenza H5N1 (n = 18).

Cat No.	Detail Description
1	Nonsuppurative encephalitis, gliosis, mononuclear infiltration in the Virchow–Robin space, vasculitis, and cerebrum congestion. A microscopic lesion in the lung was caused by severe pulmonary oedema, interstitial pneumonia, and congestion. Multifocal necrosis was found in the liver, and tubulonephritis and lymphoid depletion were present in the spleen. Influenza H5N1 diagnosed by RT-PCR.
2	At necropsy, the cat exhibited pulmonary hyperemia with multiple consolidated areas, predominantly peribronchiolar, characterised by a dark red discolouration and irregular shape. The spleen was markedly enlarged and congested. The liver was dark tan, firm, and regular in appearance. In the small intestine, numerous adult roundworms and some cestodes were found in an otherwise empty gastrointestinal tract. Influenza H5N1 diagnosed by RT-PCR.
3	The cat’s lungs were diffusely dark red with multiple coalescing yellow consolidated nodules. On cross-section, they primarily revealed dark red, irregularly shaped, peribronchiolar areas of consolidation. The bronchi and bronchioles contained abundant, tenacious yellow mucus. The pulmonary lymph nodes were moderately enlarged and moist on the cross-section. The liver contained several well-circumscribed, randomly arranged, grey to white areas measuring millimetres or up to 3 mm in diameter (necrosis). Numerous adult roundworms and some cestodes were present in the small intestine. The heart showed moderate dilation of the right ventricle. Influenza H5N1 diagnosed by RT-PCR.
4	The cat exhibited extensive postmortem loss of skin, subcutaneous tissue, and portions of skeletal muscles in the cervical and shoulder regions, likely due to scavenging. A few well-circumscribed, light brown to yellow areas, up to 2 mm in diameter, were observed in the liver, indicative of necrosis. The retropharyngeal and pulmonary lymph nodes were moderately swollen, with some ecchymosis. The lungs were severely oedematous with numerous, well-circumscribed yellow nodules. A hemohydrothorax of approximately 20 mL was observed. The mucosa of the nasal cavity, pharynx, and trachea was diffusely hyperemic. Diffuse haemorrhages were found retroperitoneally and intramuscularly in the diaphragm, within the perinephric tissue, and in the pancreas. Influenza H5N1 diagnosed by RT-PCR.
5, 6, and 7	Liver areas of necrosis with an inflammatory infiltrate composed of lymphocytes and histiocytes. Lungs, atelectasis, hyperemia, interstitial and alveolar haemorrhages. Macrophages within the alveolar lumen. Brain, perivascular infiltrates of lymphocytes and histiocytes in the white and grey matter. Perivascular infiltrate of lymphocytes and some histiocytes at the base of the choroid plexus. Bowel wall, vacuolisation and necrosis of cells in the myenteric plexus, accompanied by an inflammatory infiltrate of lymphocytes and histiocytes. Influenza H5N1 was diagnosed in cats 5 and 7 by RT-PCR.
8	Lung Interstitial pneumonia is characterised by acute vascular congestion, alveolar oedema, and fibrin exudation mixed with sparse neutrophils and macrophages into the alveolar septa and lumen. Thrombosis at the liver, multifocal random foci of acute necrosis of variable sizes (arrowheads). The necrotic areas contain fibrin, karyrhectic debris, erythrocytes, and sparse neutrophils and/or macrophages. Adrenal cortex, large foci of random acute necrosis (arrowheads). Pancreas: Severe, multifocal to coalescent necrosis manifested by extensive loss of pancreatic exocrine cells and collapse of lobules (arrowheads). An admixture of inflammatory leukocytes and karyrhectic debris is seen in the remaining stroma. The region of normal pancreatic exocrine cells is indicated for comparison with a pancreas with endothelial necrosis, fibrinoid change, and thrombosis. Influenza H5N1 diagnosed by RT-PCR.
9	The lung presented with interstitial pneumonia similar to case 1, characterised by adding more macrophages populating the alveolar septa, less evident vascular thrombosis, and some foci of mild type II pneumocyte hyperplasia. The myocardial interstitium contained some random groups of lymphocytes. The cerebral cortex presented areas of wedge-shaped lamellar malacia with variable rarefaction, cavitation, and haemorrhage. Influenza H5N1 diagnosed by RT-PCR.
10	Histologically, the brain lesions were similar to those in cases 1 and 2, with wedge-shaped areas of inflammation most frequently located in the cerebral cortex. However, the areas of glial and mononuclear cell inflammation, with mixed lymphocytes, plasma cells, and macrophages, were more extensive and numerous than those seen in previous cases, often presenting with an increased abundance of cellular debris, degenerated neurons, glial cells, and neuronal satellitosis. Influenza H5N1 diagnosed by RT-PCR.
24 and 25	Minor haemorrhages in the subcutaneous tissue over the dorsal region of the skull and multifocal meningeal haemorrhages in the brain of both cats. Microscopically, severe subacute multifocal necrotising and lymphocytic meningoencephalitis with vasculitis and neuronal necrosis, moderate subacute multifocal necrotising and lymphocytic interstitial pneumonia, moderate to severe subacute multifocal necrotising and lymphohistiocytic myocarditis, and moderate subacute multifocal lymphoplasmacytic chorioretinitis with ganglion cell necrosis and attenuation of the inner plexiform and nuclear layers. Influenza H5N1 diagnosed by RT-PCR.
28 and 29	The lung showed interstitial pneumonia, bronchiolitis, and bronchitis. The brain showed meningitis and encephalitis, while the hippocampus showed no lesions. Immunohistochemical (IHC) analysis (IL) revealed the presence of avian influenza virus (IAV) nucleoprotein in each of these organs. The brain tissue exhibited a higher level of nucleoprotein staining than the lung tissue, with a notable presence of nucleoprotein in the cerebellum and hippocampus. Influenza H5N1 diagnosed by RT-PCR.
30	Lesions in the brainstem, cerebrum, lung, and heart (subcutaneous and mild meningeal haemorrhage, mononuclear and necrotising encephalitis, and interstitial pneumonia). Influenza H5N1 diagnosed by RT-PCR.
31	Lesions in the brainstem, cerebrum, lung, heart, and eye (retina, choroid), mononuclear and necrotising encephalitis, interstitial pneumonia, and chorioretinitis. Influenza H5N1 diagnosed by RT-PCR.
32	Lesions in the brainstem, cerebellum, cerebrum, spinal cord, eye (retina, choroid), lung, tonsil, submandibular salivary gland, minor salivary glands (tongue), heart, liver, adrenal gland, and thyroid gland (mononuclear and necrotising encephalitis, interstitial pneumonia, chorioretinitis). Influenza H5N1 diagnosed by RT-PCR.
33	Lesions in the brain, eyelid conjunctiva, nasal turbinates, cribriform plate, lung, submandibular salivary gland, minor salivary glands in the pharynx, and parotid gland (mononuclear and necrotizing encephalitis, interstitial pneumonia). Influenza H5N1 diagnosed by RT-PCR.

## Data Availability

Available upon reasonable request.

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
