# Peer review of "Highly Pathogenic Avian Influenza H5N1 in Cats (*Felis catus*): A Systematic Review and Meta-Analysis"

_animals, 2025, doi:10.3390/ani15101441_

Round 1

Reviewer 1 Report

Comments and Suggestions for Authors

Dear authors, 

First of all, I have to congratulate you for this very important work. I have some questions that may improve the scientific soundness of this manuscript. I would like you not only provide responses apart for me, but also included them along the text (since my doubts may also be questions from any future reader). 

Specific questions:

1. [Line 97] Why did you not consider other databases, as Google Scholar and Scielo? Since you did not make any restriction about the language of the study, I believe it could make more sense if you also have considered more databases too.

2. [Line 266] Although the domestic cats had the highest estimated prevalence (20.0%) and considered that those category of cats are that ones which have an well-defined human tutor, I believe that this higher estimated prevalence is also due to the more feasible access to such population. I imagine that sample and/or collect data from a cat population in which has a restricted freedom and a tutor well-defined is less difficult than do it in other category of populaiton (stray, shelter, farm). Please, I would like to suggest you to think about this argumentation and to consider include it in your discussion (if it make any sense).

3. [Line 405] Considering the search criteria of this review, none studies were found from Americas. However, according to reports published by PAHO in 2025 (https://www.paho.org/sites/default/files/2025-01/2025-jan-24-phe-epiupdate-avian-influenza-eng-final.pdf) and USDA in 2024 (https://www.aphis.usda.gov/livestock-poultry-disease/avian/avian-influenza/hpai-detections/mammals), avian influenza (H5N1) virus has been detected in USA and Canada. How should the results from your systematic review be differenciated from the results showed in such reports? What comprehension do you expect the reader have from your review, since there are reports from important institutions as PAHO and USDA in which affirm that the virus has been detected in cats in Americas.

General suggestins:

1. There still are some typos in the text. Please, review carefully the writing. 

2. I believe that the expressions in Line 68 to 70 ("In these instances, ... farm environments [24, 28, 29]") and in Line 81 to 82 ("Their infections have been ... H5N1-positive cows [15, 24,39]") are very similar. It could be better to rewrite them.

3. The word "notably" appears relatively close in twice along the introduction (Line 58 and 67). I recommend an synonymous for one of them. 

Thank you for attention.

Reviewer 2 Report

Comments and Suggestions for Authors

The authors have done an excellent job of reviewing and structured the article in a complete way. Some clarifications and some very partial changes are suggested in order to make the understanding of the publication easier.

Line 63. It is mentioned that the transmission to cattle was an unexpected fact. It is suggested to insert a few lines regarding the reason why it was unexpected or the fact that, at least in theory, cattle should not have become infected and instead the jump in species took place. Is this what was meant by unexpected?

Are there any risks with pet food?

Feeding raw food (beef) to dogs and cats seems to be risky. It is suggested to emphasize these aspects if there is a real risk of transmission to the cat by this route.

Cooked food should not pose a risk but are there indications in this regard?

Tables: In particular, it is suggested to restructure the attached tables in order to make them more understandable.

Table 4. It is suggested to group the clinical manifestations on the basis of cohesive groups such as: non-specific, systemic, neurological, ocular, miscellaneous

Table 5. As above group into: e.g. Lung, hepatic, neurological, ocular, miscellaneous

Table 6. It is suggested to restructure the table by grouping the pathological pictures and indicating on the side in which cats they were found so that the reader can understand the most indicative pictures, and in a column on the side the confirmatory diagnostic method, if the authors consider it possible.

The discussion is good, easy to understand and exhaustive

Reviewer 3 Report

Comments and Suggestions for Authors

This is an excellent review of the literature on H5N1 and H5N8 in cats. I found no flaws in data presented or the conclusions drawn from that data. 

The sheer amount of data presented made the paper difficult to read but it covers the data thoroughly. Despite this criticism I am at a loss on how to minimise the data any further.

My only suggestion to improve the paper is that the authors could have included data on zoo cats, i.e. tigers, etc as there have been many reports of these species dying as a result being fed infected chicken carcasses or by contact with infected animals.
